# Phase-dependent Andreev molecules and superconducting gap closing in coherently-coupled Josephson junctions

Sadashige Matsuo [1] ✉, Takaya Imoto[1,2], Tomohiro Yokoyama[3] ✉, Yosuke Sato[1], Tyler Lindemann[4,5], Sergei Gronin[5], Geoffrey C. Gardner [5], Sho Nakosai[6], Yukio Tanaka[6], Michael J. Manfra [4,5,7,8] & Seigo Tarucha [1,9] ✉

The Josephson junction (JJ) is an essential element of superconducting (SC) devices for both fundamental and applied physics. The short-range coherent coupling of two adjacent JJs forms Andreev molecule states (AMSs), which provide a new ingredient to engineer exotic SC phenomena such as topological SC states and Andreev qubits. Here we provide tunneling spectroscopy measurements on a device consisting of two electrically controllable planar JJs sharing a single SC electrode. We discover that Andreev spectra in the coupled JJ are highly modulated from those in the single JJs and possess phase-dependent AMS features reproduced in our numerical calculation. Notably, the SC gap closing due to the AMS formation is experimentally observed. Our results help in understanding SC transport derived from the AMS and promoting the use of AMS physics to engineer topological SC states and quantum information devices.

The Josephson junction (JJ) is a representative superconducting (SC) device consisting of two weakly-linked superconductors coupled through insulators or normal conductors[1]. The JJs have been studied to engineer quantum effects in solid-state devices, enabling realization of SC quantum computing and highly sensitive magnetic sensors. The recent development of JJs based on the superconductor-semiconductor heterostructures has provided platforms for more exotic physics such as Andreev (spin) qubits[2–6], SC qubits with gate tunability[7,8] or Majorana zero modes (MZMs)[9–13]. From these aspects, engineering of coherent coupling between two JJs is an essential ingredient to explore novel SC phenomena, establish new control methods of JJs, and manage qubit operation.

Recently, the concept of short-range coherent coupling between two JJs forming Andreev molecule states (AMSs) has been proposed[14–16]. In a single JJ consisting of two superconductors and a

semiconductor, electrons are confined by the Andreev reflection at interfaces, forming the Andreev bound states (ABSs)[17–20]. In the case of two adjacent JJs sharing one SC electrode, the ABS wavefunctions in the different JJs penetrate the shared SC and overlap, which produces the coherently coupled wavefunctions called AMSs as schematically illustrated in Fig. 1a. This is an analogy of the molecular orbital states formed by the coherent coupling of two atomic states. The recent theoretical and experimental efforts on the coherently coupled JJs have revealed nonlocal SC transport derived from the AMSs such as the nonlocal Josephson effect[21–24] and the Josephson diode effect[25–27]. Additionally, the AMS physics in the coupled JJs may provide novel insights on the SC transport intermediated by the Cooper pair splitting in a parallel double quantum dot or double nanowire JJ, which can be regarded as the two JJs sharing two SC electrodes[28–32]. In order to understand the microscopic mechanisms of such SC transport for the

[1]Center for Emergent Matter Science, RIKEN, Saitama 351-0198, Japan. [2]Department of Applied Physics, Tokyo University of Science, Tokyo 162-8601, Japan. [3]Department of Materials Engineering Science, Osaka University, Osaka 560-8531, Japan. [4]Department of Physics and Astronomy, Purdue University, West Lafayette, Indiana, IN 47907, USA. [5]Birck Nanotechnology Center, Purdue University, West Lafayette, Indiana, IN 47907, USA. [6]Department of Applied Physics, Nagoya University, Nagoya 464-8603, Japan. [7]School of Materials Engineering, Purdue University, West Lafayette, Indiana, IN 47907, USA. [8]Elmore Family School of Electrical and Computer Engineering, Purdue University, West Lafayette, Indiana, IN 47907, USA. [9]RIKEN Center for Quantum Computing, RIKEN, Saitama 351-0198, Japan. ✉e-mail: sadashige.matsuo@riken.jp; tomohiro.yokoyama@mp.es.osaka-u.ac.jp; tarucha@riken.jp

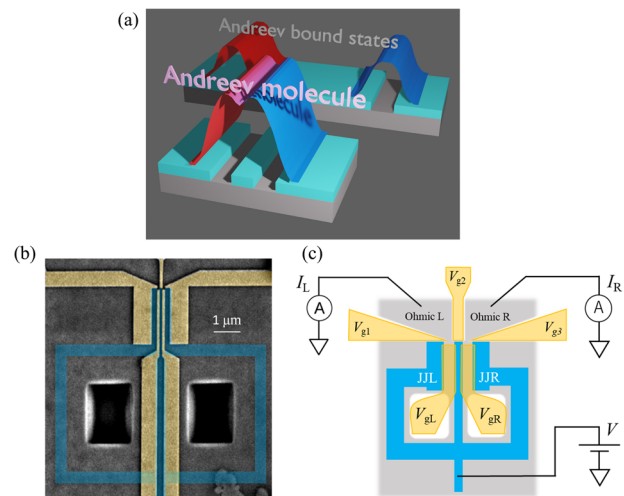

**Fig. 1 | A device for tunneling spectroscopy of the coupled JJs. a** A schematic illustration of the AMS in the coupled JJ. **b** A scanning electron microscope image of the coupled planar JJ device. The blue and yellow regions represent the SC electrodes and the gate electrodes, respectively. Two JJs named JJL and JJR are coupled through a shared SC electrode and each is embedded in an SC loop. **c** A schematic image of the device. Tunnel currents through QPCL and QPCR are measured, and from the result, the differential conductance is calculated for each to implement the tunneling spectroscopy of JJL and JJR.

novel SC device functionality and engineer operation and integration of the Andreev qubits using the coupled JJs[33], observation of the AMSs in the coupled JJs is indispensable. However, experimental evidence of the AMS formation in the coupled JJs is lacking although there are experimental reports of the AMS signatures formed in SC junctions other than JJs[34–36].

The ABS spectra of two-terminal JJs have intensively been studied. For example, the ABS spectra of short ballistic JJs do not depend on whether the JJs hold the spin-orbit interactions (SOIs) or not[37–39] and the SC gap closes only when the junction transmission is unity[17]. In the long JJ case, the spin-split ABSs can appear in the spectra due to the SOIs[40]. When the spin-split ABSs cross at zero energy and the SC gap closes, various exotic SC phenomena including MZMs emerge. For example, the topological transition is induced by phase control at the SC gap closing points in the two-terminal planar JJs in the presence of a Zeeman filed and the MZMs appear[9–12,41].

Multiterminal JJ structures consisting of three or more SC electrodes contacted on a single normal metal[42–52] are proposed to generate the Andreev spectra drastically modulated from those of the two-terminal JJs. Especially in the presence of a strong SOI, the large splitting of Kramers degeneracy by phase control is proposed to provide the spin-split ABSs' crossing at zero energy regarded as the SC gap closing[42,51]. Related experimental results have been reported[53]. The SC gap closing especially in the coupled planar JJs, which is regarded as a kind of the multiterminal JJ causes the topological transition and emergence of MZMs by phase control with no Zeeman effect[54,55]. Therefore, a search for the SC gap closing in coupled JJs and understanding of the mechanism are important for topological SC research.

## Concept and device of this study

Here, we experimentally study the Andreev spectrum in single and coupled JJs by tunneling spectroscopy[11,56] to elucidate the phase-dependent AMSs. For this purpose, we fabricate an SC device consisting of two JJs named JJL and JJR sharing an SC electrode from a high-quality InAs quantum well covered with the epitaxially grown thin Aluminum layer[57–59]. We note that the InAs quantum well possesses strong SOI. A scanning electron microscopic image and the schematic image of the coupled JJ device are shown in Fig. 1b, c, respectively. The

separation between the two JJs corresponding to the width of the shared SC electrode is designed as 150 nm which is sufficiently shorter than the coherence length of Aluminum (Al). The junction length and width are 80 nm and 1.9 μm, respectively. The two JJs are each respectively embedded in a SC loop which encloses the same area, to induce the same phase difference in JJL and JJR. With definition of $\phi_L$ and $\phi_R$ of the phase differences on JJL and JJR as $\phi_L = \theta_S - \theta_l$ and $\phi_R = \theta_r - \theta_S$, respectively, the out-of-plane magnetic field $B$ changes the phase differences with $\phi_L = \phi_R$. Here $\theta_l$, $\theta_S$, and $\theta_r$ represent the phases of the left, shared, and right SC electrodes, respectively. The main concept of this design is to compare the spectroscopic results in the single JJ and the coupled JJ cases using the same device but by electrically independently controlling the two JJs. For this purpose, we have fabricated the gate electrodes as highlighted in yellow in Fig. 1b. The gate electrodes on JJL and JJR are used to control the planar JJs with the gate voltages of $V_{gL}$ and $V_{gR}$, respectively. Additionally, three gate electrodes are prepared to electrically form the quantum point contacts on the edges of the planar JJs (see supplementary Note 2). The gate voltages $V_{g1}, V_{g2}$, and $V_{g3}$ are applied on the electrodes as depicted in Fig. 1c. We name the point contact formed on the edge of JJL (JJR) as QPCL (QPCR). When performing the spectroscopy of JJL, we pinch off the QPCR and detect the tunnel current through the QPCL as shown in the schematic circuit diagram in Fig. 1c. We measure this device at 10 mK of the base temperature in our dilution refrigerator. For the tunneling spectroscopy, we apply a D.C. bias voltage $V$ with a small oscillation component (5 μV) and measure the tunnel currents through QPCL and QPCR by Lock-in amplifiers. Then, we obtain $G_L$ and $G_R$, the differential conductance through the QPCL and QPCR, respectively.

## Results

### Tunneling spectroscopy of the single JJs

First, we perform the tunneling spectroscopy of the single JJs. Initially, we measure $G_L$ with JJR pinched off ($V_{gL}, V_{gR}$) = (0 V, −6 V). In the measurement, QPCR is pinched off also and QPCL is tuned to allow $G_L \sim 0.10 e^2/h$. Then the obtained $G_L$ as a function of $V$ and $B$ is indicated in Fig. 2a. As clearly seen, the subgap structure emerges inside the Aluminum SC gap energy (-0.18 meV). The observed SC gap inside the Aluminum gap energy is modulated periodically as a function of $B$. The period is around 0.126 mT consistent with the expected period evaluated from the loop area (0.172 mT). This periodic modulation has been reported in previous studies[11,19,56] and assigned to the phase modulation of ABSs in the single ballistic JJ[17,56]. Therefore, the oscillation period of the gap is equivalent to $2\pi$ and $B$ producing the maximum (minimum) gap corresponds to $\phi_L = 0 \pmod{2\pi}$ ($\phi_L = \pi \pmod{\pi}$). It is noted that JJR is pinched off so that $\phi_R$ is not considered here (see Supplementary Notes 3 and 4, and Figs. S3 and S5). In our results, the maximum (minimum) SC gap is around 0.1 meV (0.08 meV). The energy of each ABS depends both on the phase differences and on the phase accumulated by the quasiparticles traversing the JJs at various angles from the junction-width direction. Therefore, their energies disperse to form a continuous sub-gap spectrum[12]. These results imply that our method of tunneling spectroscopy using QPCL correctly detects the Andreev spectrum in JJL. We note that the jump of the data at $B = 0.066$ mT occurs because of the charge jump around QPCL.

We next perform characterization of JJR with ($V_{gL}, V_{gR}$) = (−6 V, 0 V). We measure $G_R$ as a function of $V$ and $B$ with QPCL and JJL pinched off. The results are shown in Fig. 2b. As with the JJL results, the periodic oscillation of the SC gap is observed, which is the ABS oscillation of the single JJR as a function of $\phi_R$. In the JJR results, the maximum (minimum) SC gap is around 0.12 meV (0.075 meV). Comparing the JJR result with the JJL result, the oscillation periods in both are consistent, which reflects the two SC loops have the same area. Therefore, the results imply that our device works correctly to detect the ABSs of both the single planar JJs with the phase biased by the out-of-plane magnetic field.

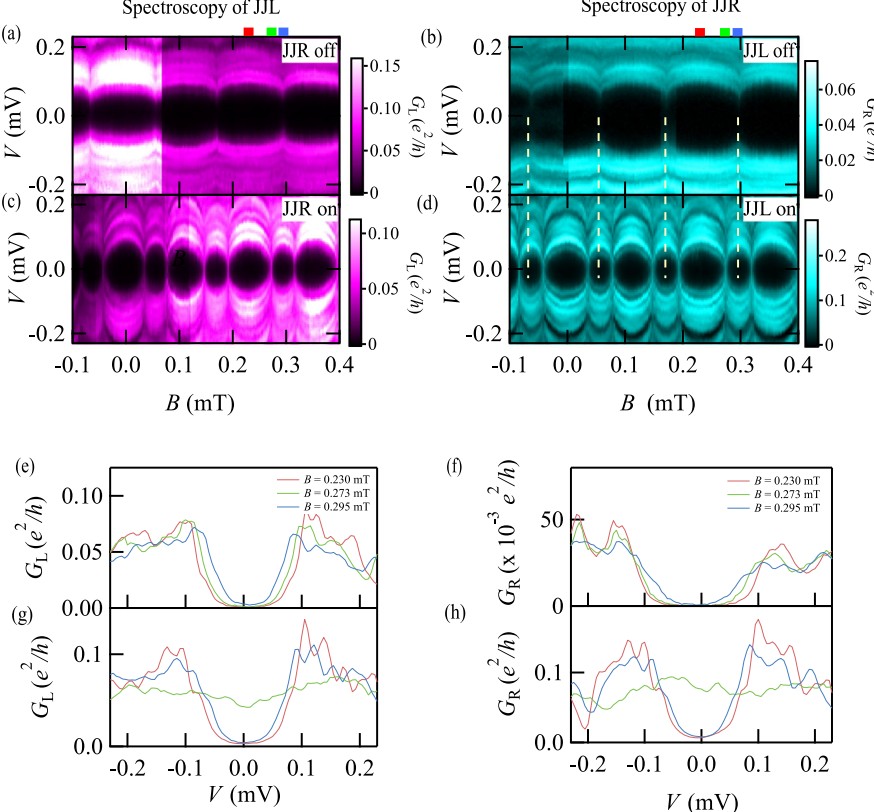

**Fig. 2 | Tunneling spectroscopy results of the single and coupled JJs. a** Tunneling spectroscopic result of the single JJL as a function of $B$. The SC gap oscillates to B, which is assigned to the feature of ABS expected in short JJs. **b** Tunneling spectroscopic result of the single JJR. Almost the same features as in the JJL result are found. The periodicity of the ABS oscillation is the same as that in (**a**) because the two loops hold the same area. The white dashed lines indicate the $B$ points giving the minimal SC gap. The red, blue, and green tags on the top axes of (**a**) and (**b**) indicate $B = 0.230$, 0.273, and 0.295 mT, respectively. **c** Tunneling spectroscopic result of the JJL coupled with JJR. The result is drastically modulated from the single JJ case in (**a**). Especially, the SC gap becomes minimal away from $\phi_L = \phi_R = \pi$ where

the gap becomes minimal in the single JJ cases. **d** Tunneling spectroscopic result of the JJR coupled with JJL. The same features observed in (**c**) were acquired. The consistency between (**c**) and (**d**) assures that the AMSs are constructed due to the coherent coupling of the two JJs. **e** $G_L$ vs. $V$ at $B = 0.230$, 0.273, and 0.295 mT, respectively, measured for the single JJL. **f** $G_R$ vs. $V$ measured for the single JJR. **g** $G_L$ vs. $V$ measured for the coupled JJL. The SC gap is closed at $B = 0.273$ mT while the other two curves hold the SC gap. **h** $G_R$ vs. $V$ measured for the coupled JJR. As with the coupled JJL case, the SC gap is closed at $B = 0.273$ mT while the other two curves hold the SC gap.

## Tunneling spectroscopy of the coupled JJs

Next, we explore the tunneling spectroscopy of JJL with JJR on $(V_{gL}, V_{gR}) = (0V, 0V)$. Here two JJs are turned on and then both of $\phi_L$ and $\phi_R$ evolve as a function of $B$ with $\phi_L = \phi_R$ assured by the same SC gap oscillation period of the single JJL and JJR results. The obtained spectroscopic result ($G_L$ as a function of $B$ and $V$) is shown in Fig. 2c. Compared with the single JJ result in Fig. 2a, the Andreev spectrum is drastically modulated. This drastic change is invoked by turning on JJR, namely turning on the coherent coupling between the two JJs. Therefore, the observed spectroscopic result and the change from the single JJ results are assigned to the formation of AMSs in the coupled JJs. Here a significant feature that cannot be found in the single JJ results is observed. In the coupled JJ case, the SC gap becomes maximal where the gap in the single JJL result in Fig. 2a produces the minimum. This means that $\phi_L = \phi_R = 0, \pi \pmod{2\pi}$ give the maximal SC gap in Fig. 2c and the minimum SC gap in Fig. 2c is realized in $0 < \phi_L = \phi_R < \pi$.

When the coherent coupling between JJL and JJR modulates the JJL Andreev spectrum, the JJR spectrum should also be modulated. We confirm modulation of the JJR spectrum by the tunneling spectroscopy of JJR with $(V_{gL}, V_{gR}) = (0 \text{ V}, 0 \text{ V})$. The obtained $G_R$ as a function of $B$ and $V$ is shown in Fig. 2d. As explicitly found, the same feature as in the JJL results appears. The consistency obtained between the JJL result in Fig. 2c and the JJR result in Fig. 2d assures that the Andreev spectrum modulation is induced by the coherent coupling between the two

planar JJs. We note that these features are reproducible (see Supplementary Note 1 and Figs. S1 and S2). Particularly, both results exhibit the same dependence on the phase differences, which means that the coherent coupling maintains the phase coherence.

To evaluate the minimum SC gap energies in the coherent coupling case, we extract and plot the line profiles at $B = 0.230$, 0.273, and 0.295 mT, in Fig. 2a–d in red, green, and blue in Fig. 2e–h, respectively. Note that the SC gap changes from the maximum to the minimum in this $B$ field range in the single JJ, while from the maximum to the minimum and then to the maximum in the coupled JJs. Figure 2e, f indicate the line profiles of the single JJL and JJR results in Fig. 2a, b, respectively. As seen in Fig. 2e, f, the Andreev spectrum is always gapped. This holds at any $B$ field. On the other hand, in Fig. 2g, h, the green line profile does not touch the $G = 0$ line. Therefore, the SC gap looks closed in our setup resolution (~0.01 mV). Note that the same result is obtained for each SC gap minimum.

## Discussion
### Numerical calculation of the AMS and Discussion of super-conducting gap closing

To reveal that the modulated Andreev spectrum reflects the AMS properties owing to the coherent coupling through the shared SC electrode, we perform numerical calculations of Andreev spectra of the single and coupled JJs using a tight-binding model (see

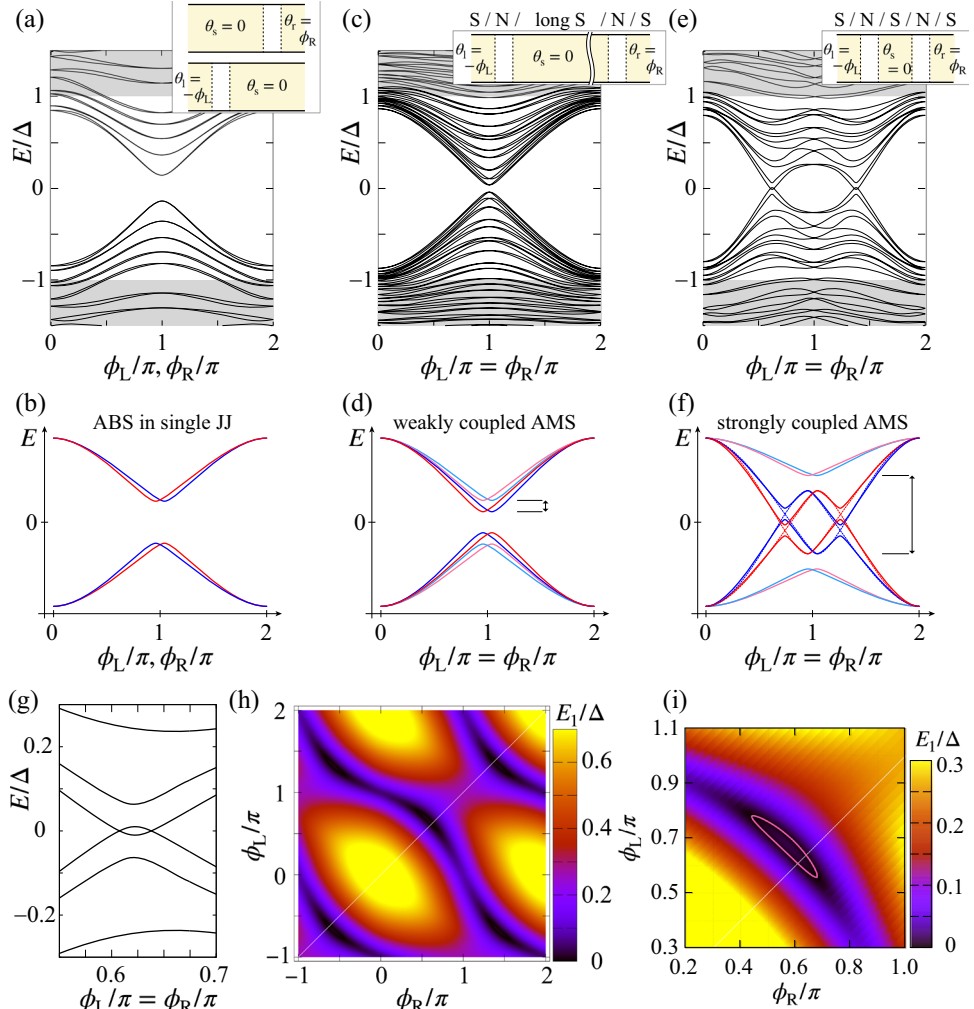

**Fig. 3 | The numerical calculation results indicating the AMS formation and SC gap closing. a** The calculated Andreev spectrum of a single JJ with the SOI. The inset is a schematic image of the considered single JJ (isolated JJL or JJR). The yellow and white regions represent the SC electrode and the normal region, respectively. The phases of the SC electrodes are written. **b** A conceptual image of the single JJ's Andreev spectrum with the SOI. The SOI makes the spin degenerated ABSs split into the blue and red ABS curves. **c** The calculated Andreev spectrum of a weakly coupled JJ with the SOI through the 1000 nm-length central SC electrode in $0 \leq \phi_L = \phi_R \leq 2\pi$. The SC gap becomes smaller than that in (**a**). **d** A conceptual image of the weakly coupled JJ's spectrum. Due to the AMS formation, hybridization of the red (blue) ABSs in JJL and JJR in (**b**) invokes level splitting into the bonding and anti-bonding AMSs highlighted with the dark red and pink (the dark blue and cyan).

Consequently, the SC gap becomes smaller. The arrow represents the level repulsion due to the hybridization to form the AMSs at the local minimum. **e** The calculated Andreev spectrum of a strongly coupled JJ with the SOI through the 160 nm-length center SC electrode. The SC gap becomes local maximal at $\phi_L = \phi_R = \pi$ and the SC gap closing is obtained in $0 < \phi_L = \phi_R < \pi$ and $\pi < \phi_L = \phi_R < 2\pi$. **f** A conceptual image of the strongly coupled JJ's spectrum. Strong coupling produces the level crossing of the dark blue and dark red AMSs at zero energy, corresponding to the SC gap closing. The arrow represents the same level repulsion as shown in (**d**). **g** The enlarged view around the SC gap closing in (**e**) around $\phi_L = \phi_R = 0.63\pi$. **h** The energy of the positive lowest AMSs ($E_1/\Delta$) as a function of $\phi_L$ and $\phi_R$. The white line indicates the condition of $\phi_L = \phi_R$ corresponding to the experimental situation. **i** The enlarged view of the area surrounded by the zero energy states.

Supplementary Note 7 and Fig. S7). The calculation includes the strong Rashba-type SOI in the normal region expected in the InAs quantum well. To explain the mechanism to generate the AMS, we numerically calculate the Andreev spectra in three cases: isolated single JJs, a coupled JJ through a long central SC electrode (a weakly coupled JJ), and a coupled JJ through a short central SC electrode (a strongly coupled JJ) in Fig. 3a, c, e, respectively. The inset in each figure shows a schematic image of the JJ structure consisting of SC regions in yellow and normal regions in white. The spectrum of the isolated JJL or JJR in Fig. 3a reflects a typical behavior of the Andreev spectrum in a single JJ in which the SOI causes a weak spin splitting of the ABSs and the SC gap is open around $\phi_L, \phi_R = \pi$ with imperfect transmission[40] as schematically depicted in Fig. 3b. The red and blue ABSs in Fig. 3b represent the SOI-induced spin splitting of ABSs as time-reversal invariant pairs.

When the two JJs (JJL and JJR) share the long SC electrode with a length of 1000 nm, which is comparable to the coherence length of

Aluminum, the coherent coupling of the ABSs is weak. The calculated spectrum indicates the smaller SC gap than that of the isolated single JJs. This is a consequence of the AMS formation. The coherent JJ coupling forms the bonding and anti-bonding AMSs due to the hybridization of spin-split ABSs in the respective JJs[14–16]. Figure 3d depicts the schematic image. The red ABSs of JJL and JJR in Fig. 3b form the bonding (dark red) and anti-bonding (pink) AMSs in Fig. 3d and the blue ABSs in Fig. 3b form the dark blue and cyan AMSs in Fig. 3d. Therefore, the AMS formation causes a reduction of the SC gap because the dark red and blue AMSs of the positive (negative) energies are pushed down (up) from the ABSs in the isolated single JJs in the presence of the coherent coupling.

When the center SC electrode is shortened to 160 nm length, the coherent coupling becomes more significant and the calculated result in Fig. 3e exhibits that the SC gap is maximal at $\phi_L = \phi_R = \pi$ and the SC gap becomes minimal in $0 < \phi_L = \phi_R < \pi$ and $\pi < \phi_L = \phi_R < 2\pi$. Figure 3g

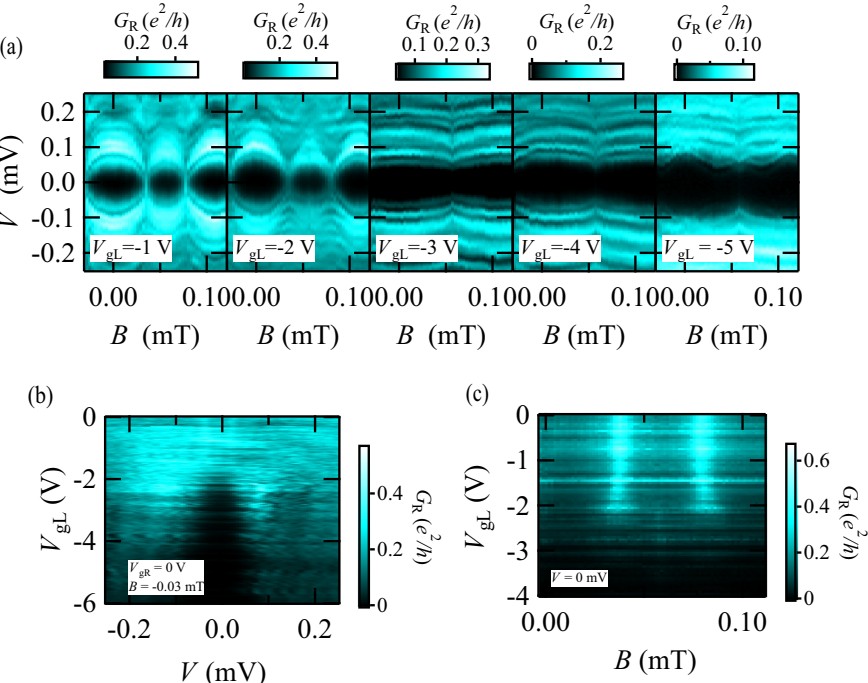

**Fig. 4 | Nonlocal gate control of the AMSs detected in the coupled JJR. a** $G_R$ as a function of $B$ and $V$ at several $V_{gL}$. The AMS features appear for $V_{gL} \geq -2$ V but disappear for $V_{gL} < -2$ V. This suggests that the JJL is pinched off at $V_{gL} \sim -2$ V. **b** $G_R$ as a function of $V$ and $V_{gL}$ at $V_{gR} = 0$ V and $B = -0.03$ mT. $B = -0.03$ mT corresponds to the SC gap closing point. Then the result indicates no gap structure for

$V_{gL} \geq -2$ V while the SC gap grows as $V_{gL}$ becomes smaller than $-2$ V. This indicates that the SC gap closing is robust as long as the AMS is formed. **c** $G_R$ as a function of $B$ and $V_{gL}$ at $V = 0$ mV. The two vertical bright lines represent $B$ producing the SC gap closing. The lines disappear around $V_{gL} \sim -2$ V, supporting that the SC gap closing is robust as long as the AMS is formed.

is an enlarged view around the minimal SC gap in Fig. 3e, indicating the SC gap closing occurs at two points. These features are explained in Fig. 3f. Due to the strong coherent coupling, the level splitting becomes larger than the SC gap in the isolated single JJs in Fig. 3b. Then, the positive (negative) bonding AMSs go down to negative energy (up to positive energy) around $\phi_L = \phi_R \simeq \pi$ as the dark red and dark blue dashed curves in Fig. 3f. Owing to the spin splitting of the AMSs, level crossing of the AMSs with the same colors does not occur at zero energy while the level crossing of the AMSs with the different colors occurs at zero energy. At the crossing points of the same colored dashed AMSs, level hybridization modulates the spectrum to the solid curves. On the other hand, the crossing of the AMSs with the different colors is protected because the two AMSs with the different colors are time-reversal invariant pairs (the Kramers pairs) (see Supplementary Notes 9 and 11 and Figs. S9 and S11). Therefore, when the spin splitting is sufficiently large compared to the level hybridization of the same colored AMSs, the AMS crossing at zero energy, namely the SC gap closing emerges.

The numerical results shown in Fig. 3e and the schematic image in Fig. 3f explain the properties of the experimentally observed spectra in Fig. 2. For example, the SC gap becomes minimal in the middle of $0 < \phi_L = \phi_R < \pi$ and then $\phi_L = \phi_R = 0, \pi$ indicates local maxima. Our experimental data indicates the gap closing at single points in $0 < \phi_L = \phi_R < \pi$ while the numerical results predict the gap closing at two points related to the splitting of the Kramers degeneracy. This discrepancy between the experimental results and theoretical calculation can be attributed to the effect of our measurement resolution and thermal smearing. Due to the effect, the two SC gap closing points in the strong SOI case depicted in Fig. 3e can overlap or the small SC gap in the weak or negligible SOI case (Supplementary Note 11 and Fig. S11) is smeared out. In any case, the experimentally observed Andreev spectra in Fig. 2 reflect the phase-dependent AMSs between the two JJs.

The calculated results depend on the samples for the on-site random potential. For the strength of randomness corresponding to a 217 nm mean free path in agreement with our InAs quantum well, we obtain the gap closing in the AMS spectrum in 19% of samples in the simulation (see Supplementary Notes 7 and 8 and Figs. S7 and S8).

Figure 3h exhibits the positive lowest energy of the Andreev spectra in the plane of $\phi_L$ and $\phi_R$. A dark region in the vicinity of $\phi_L = \phi_R = 0.63\pi$ exhibits a finite area surrounded by the zero energy states corresponding to the SC gap closing points as found in the enlarged view in Fig. 3i. The emergence of the zero energy states in Fig. 3h, i has been discussed in terms of the SC phase winding in the multiterminal JJs holding the SOIs[42,51]. The theory[42] proposes that the zero energy states can appear in $\pi \leq \phi_L + \phi_R \leq 2\pi$ with $\phi_L, \phi_R \in [0, \pi]$ and $2\pi \leq \phi_L + \phi_R \leq 3\pi$ with $\phi_L, \phi_R \in [\pi, 2\pi]$ in our definition (see Supplementary Note 10 and Fig. S10). The experimental and numerical SC gap closing is obtained in $\pi/2 < \phi_L = \phi_R < 3\pi/2$ at the experimentally and numerically obtained SC gap points, satisfying the condition. The SC gap closing and the zero energy states mean that the Kramers degeneracy is lifted in the coupled JJs with the strong SOIs. It is theoretically discussed that the SC gap closing by lifting the Kramers degeneracy in the coupled planar JJs holding the strong SOI is related to the topological transition[54,55] (to the topological symmetry class D of the Altland-Zirnbauer classification[60–63]). Following the prediction, the gapped region surrounded by the zero energy states in Fig. 3i can be the topological SC state. In this context, our finding of the SC gap closing in the coupled JJs might be related to the topological SC states in the coupled JJs.

## Nonlocal gate control of the AMS
Finally, we explore the gate voltage dependence of the AMSs and the SC gap closing. We focus especially on the nonlocal gate control effect

on the AMS results. Figure 4a shows $G_R$ as a function of $B$ and $V$ at $V_{gL} = -1, -2, -3, -4$, and $-5$ V with $V_{gR} = 0$ V. It is clear that the AMS features disappear and the ABSs in the single JJR emerge for $V_{gL} \leq -3$ V. Then we fix $B = -0.03$ mT at the SC gap closing and sweep $V_{gL}$ as shown in Fig. 4b. The gap-closing behavior appears robust for $V_{gL}$ from 0 to $-2$ V and gapped for $V_{gL} \lesssim -2$V. With Fig. 4a, this indicates that JJL is pinched off around $V_{gL} = -2$ V and the SC gap closing disappears when the AMSs disappear. This is supported by the magnetic field dependence of $G_R$ measured as a function of $V_{gL}$ shown in Fig. 4c. The two bright vertical lines are observed in the gate voltage range of $-2$ V $< V_{gL} \leq 0$ V. This indicates the SC gap closing of JJR occurs within $-2$ V $< V_{gL} \leq 0$ V and almost at the same magnetic fields in the nonlocal gate control. Similar behavior is also observed in the nonlocal gate control of the JJL (see Supplementary Note 5 and Figs. S4 and S5). Furthermore, we confirm that the SC gap closing is robust against the control of $V_{gR}$ as long as the AMSs are formed (see Supplementary Note 6 and Fig. S6). These results reveal that the AMSs are formed when both JJL and JJR exist and are coherently coupled with each other even if their carrier densities are different. In addition, the SC gap closing is robust against the control of the carrier densities of JJL and JJR. On the other hand, the numerical calculation does not always produce the SC gap closing in the spectrum. This difference may be related to a hidden physical mechanism in the coherently coupled JJs or the finite energy resolution of the tunneling spectroscopy of our setup. Further studies are demanded. We found that similar spectroscopic results in a three-terminal JJ on an InAs quantum well have been reported on arXiv[64] while preparing this manuscript.

## Summary

As a summary, we perform the tunneling spectroscopy of the electrically controllable planar JJs embedded in the two SC loops. We find that the Andreev spectrum of single JJs indicates the periodic ABS oscillation as expected in ballistic JJs while the coupled JJs exhibit the strongly modulated structures indicating the SC gap closing. The modulated Andreev spectrum is discovered in both of the JJs and is reproduced by theoretical calculation. From these results, it is concluded that the AMSs in the coupled planar JJs are detected. This study will contribute to elucidating the microscopic mechanisms of the AMS formation and developing quantum control and integration of Andreev qubits. Additionally, our results suggest that the coupled planar JJs will provide a new platform to engineer the MZMs using only phase control.

## Methods

### Sample growth

The wafer structure has been grown on a semi-insulating InP substrate by molecular beam epitaxy. The stack materials from the bottom to top are a 100 nm $In_{0.52}Al_{0.48}As$ buffer, a 5 period 2.5 nm $In_{0.53}Ga_{0.47}As$/ 2.5 nm $In_{0.52}Al_{0.48}As$ superlattice, a 1 μm thick metamorphic graded buffer stepped from $In_{0.52}Al_{0.48}As$ to $In_{0.84}Al_{0.16}As$, a 33 nm graded $In_{0.84}Al_{0.16}As$ to $In_{0.81}Al_{0.19}As$ layer, a 25 nm $In_{0.81}Al_{0.19}As$ layer, a 4 nm $In_{0.81}Ga_{0.19}As$ lower barrier, a 5 nm InAs quantum well, a 10 nm $In_{0.81}Ga_{0.19}As$ top barrier, two monolayers of GaAs and finally an 8.7 nm layer of epitaxial Al. The top Al layer has been grown in the same chamber without breaking the vacuum. The two monolayers of GaAs are introduced to help passivate the wafer surface where the Al film is removed and to make the sample more compatible with the Al etchant, which does not attack GaAs. The two-dimensional electron gas (2DEG) is accumulated in the InAs quantum well.

### Device fabrication

In the fabrication process, we have performed wet etching of the unnecessary epitaxial Aluminum with a type-D aluminum etchant to form JJs and SC loops. Then, we have grown a 30 nm-thick aluminum oxide layer through atomic layer deposition and fabricated a separate gate electrode on each junction with 5 nm-Ti and 20 nm-Au.

## Data availability

The data that support the findings of this study are available on the Zenodo repository at https://doi.org/10.5281/zenodo.8327633.

## Code availability

All the codes used in this work are available upon request to the authors.

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

## Acknowledgements

We thank Dr. Russell Deacon for his careful reading and valuable comments. This work was partially supported by a JSPS Grant-in-Aid for Scientific Research (S) (Grant No. JP19H05610), JST PRESTO (grant no. JPMJPR18L8), JST FOREST (Grant No. JPMJFR223A), Advanced Technology Institute Research Grants, and JSPS Grant-in-Aid for Early-Career Scientists (Grant No. 18K13484).

## Author contributions

S.M. conceived the experiments. T.L., S.G., G.C.G., and M.J.M. grew the quantum well wafer. S.M. fabricated and measured the device with T.I.; S.M., T.I., Y.S., and S.T. analyzed the data; S.N. and Y.T. gave the theoretical input. T.Y. performed the numerical calculation. S.T. supervised the study.

## Competing interests

The authors declare no competing interests.
