## [Peer Review File · Nature Communications]

Phase-dependent Andreev molecules and superconducting gap closing in coherently coupled Josephson junctionsREVIEWER COMMENTS

Reviewer #1 (Remarks to the Author):

REPORT on "Phase-dependent Andreev molecules and superconducting gap closing in coherently coupled Josephson junctions" (NCOMMS-23-19143-T) by Dr Matsuo

In the manuscript the authors address an interesting topic : an artificial molecule resulting from two coupled Josephson junctions. The Andreev bound states in each junction play the role of the energy levels of an atom while the overlap among ABS from two different junctions sharing a common electrode changes the energy spectrum of each junction and simulates a molecule. This result is not very surprising by itself but it opens up new perspectives in the field of multi-terminal Josephson junctions which has been recently suggested as an interesting platform for topological matter.

The junctions are obtained by coupling two aluminium superconducting electrodes through a In As quantum well. The experiment is performed with care and the results sound correct.

The ABS energy depends on the phase difference. In this work, phase bias of the two Josephson junctions is provided by the magnetic flux through two identical superconducting loops. By tunneling spectroscopy, it is shown that when an Andreev molecule forms then the gap closes as a function of the applied magnetic flux. This is the main result. Simulations seem to indicate that this is the case only when spin-orbit coupling (SOC) is included. However the role of SOC in gap closing is unclear. For instance, no simulations are shown for a single junction.

Without a clear and convincing explanation on why SOC is essential for gap closing I can not recommend publication. A quasi quantitative analysis supporting such a conclusion is also needed. This is important also because recent work on very similar devices (arXiv:2302.14535) seems not to show such a gap closing. On the same foot it would be interesting to show that the gap closing disappears when the width of the electrode common to both Josephson junctions becomes larger than the superconducting coherence length.

Finally a more focused introduction on the consequences of the SOC on the ABS spectrum would help the reader in a better understanding of the main result. Some sentences (see for instance : « the induced SC gap is defined as the ABSs with no momentum along the junction and the ABSs with finite momentum fill the states outer of the gap ») sound unclear.

Reviewer #2 (Remarks to the Author):

In "Phase-dependent Andreev molecules and superconducting gap closing in coherently coupled Josephson junctions" Matsuo et. al report tunnel spectroscopy of two coupled Josephson junctions. The authors perform individual spectroscopy of each junction with the other one depleted, effectively realizing an ordinary two-terminal junction, and then with both junctions activated to study the coupled case. Some modeling of spin-orbit coupled Andreev bound states is provided. The authors show how changing the gate voltage on one junction continuously alters the spectroscopy in another junction. Particular attention is paid to an experimentally observed gap closure feature.

The main claims are that the two junctions are coherently coupled and that the superconducting gap closes due to the formation of an Andreev molecule. It does appear that there is a coherent interaction between Andreev states in the two junctions. However, I am concerned the discussion surrounding the gap closure. In particular, the authors relate their observation of a gap closure to the presence of spin-orbit interaction in the junction. I do not think this claim is actually supported by the data.

First, I would like to emphasize that as an experimental observation, the gap closure is completely clear, as shown in Fig.'s 2g,2h. I have no problem with this. The problem lies with the discussion of these features and their interpretations.

Although the authors state "Observation of this SC gap closing is an important step to realize and control more exotic SC phenomena predicted in the multiterminal JJs including Weyl fermion physics" I do not see a completely clear claim in the main text as to the origin of the gap closure. The authors show some numerical simulations and discuss related theories of topological transitions in planar multi-terminal junctions or time-reversal symmetry breaking in small multi-terminal junctions.

My reading is that the authors are saying that, based on comparing the data with numerical simulations in Fig. 3a,3b, the data support the spin-orbit coupled case in 3b. However, I do not believe the data in reality distinguish between the small gap scenario (3a) and "gapless" scenario (3b). I use quotes because actually both cases are gapped in the provided simulations. It would indeed be interesting if there was evidence that spin-orbit is making the gap close, but the data the authors present appears entirely consistent with both cases.

A related point of confusion is what regime the authors actually claim this gap closure corresponds too. Since the fluxes are equal I presume they do not intend to claim time-reversal symmetry breaking in the junctions. Is the idea that this is akin to the fine-tuned class DIII point of Ref. [30], which is indeed gapless after ensemble averaging?

I am not enthusiastic about this manuscript appearing in nature communications, but if the authors wish to respond I would like to see a much more clear statement about what claims are being made surrounding the gap closure, and a careful argument of why the data agree better with 3(b) than 3(a).

If I take away the discussion of gap closure, there remains perfectly fine evidence that an Andreev molecule has been realized. But, as the authors point out in their literature review, other Andreev molecules have also appeared, and this case does not appear particularly clean or tunable.

As a minor comment, in the Fig. 1(h) caption "GR vs. V measured for the coupled JJL." I think the authors intend JJL -> JJR?

Reviewer #3 (Remarks to the Author):

I have reviewed the manuscript entitled "Phase-dependent Andreev molecules and superconducting gap closing in coherently coupled Josephson junctions" which reports primarily experimental results on tunneling spectroscopy in a three-terminal setup comprising two coupled Josephson junctions. The reported experimental findings appear to be solid and reproducible with relevance for the readers of Nature Communications. I have, however, two reservations to be addressed before I can recommend the manuscript for publication.

First, I find the language of the manuscript somewhat clumsy with parts which are not really clear such as, e.g., the sentence "The induced gap..." on lines 120 and 121 or "oscillates TO B" on l. 422. General check of English would be beneficial. Furthermore, I would recommend to explicitly mention in the caption of Fig. 2 the three colors above the upper panels corresponding to the lines in the lower panels. This is mentioned in the main text only and is thus somewhat missing in the caption itself. My last question about the presentation concerns the Fig. 4. Is really the mag. field axis in panel 4c 10-times longer than in panels 4a? The gap closing within the range of 0.1 mT of panels 4a is not seen at all in panel 4c - is it deliberate and, if so, why?

More serious problems I see in the modelling part of the manuscript. Even though I do understand a very short space for any detailed description of the theoretical concepts, I haven't got any specific image of the mechanism involved in the observed effects. The whole section on "Numerical calculation of the AMS..." is just a series of vague hints on various phenomena and list of citations without any more specific conclusion (in my view). On the other hand, the calculations in the supplement 7 are for a very specific model without any context (no references at all). Altogether, neither the main text nor the supplementary note provide a coherent and/or conclusive picture of the expected microscopic mechanism, which I view as a serious shortcoming of the present manuscript. I strongly suggest the authors to amend this situation.

Revision List

We uploaded a supporting file (main_v9_compared) in which we highlighted the revisions with red and indicated the line numbers.

#	Revision on the main text	How to revise
1	L20-29	We refined our abstract to reduce the word count in 150 words.
2	L56-62	We added an explanation of background about the SOI roles in two-terminal JJs' Andreev spectra.
3	L63-71	We completely refined our explanation of the multiterminal JJs' spectra.
4	L77-78	We added a sentence of our InAs quantum well holding the SOI.
5	L84-85	We changed the characters to represent the SC phases from ϕ_i ($i = l, s, r$) to θ_i .
6	L114-116	We changed our explanation of the continuous subgap spectrum with the proper reference.
7	L164-L226	We completely improved our discussion with the numerical calculation.
8	L261	We removed our future prospect of "Weyl singularities".
9	L454	We added an explanation of the red, blue, and green tags.
10	Fig. 3	We replaced Fig. 3 and the caption.
11	Fig. 4	We corrected our mistake on the label.
12	L415	We added our new acknowledgement to Dr. Deacon who polished our English.
13		We corrected our English phrasing and some typos. We added some references in the list.

We uploaded a supporting file (supplementary_v7_compared) in which we highlighted the revisions with red and indicated the line numbers.

#	Revision on the supplemental material	How to revise
14	Supplementary note 7	We improved our explanation of the model. Especially, we explained the complete Hamiltonian and how to use it for our calculation.
15	Supplementary note 8	We improved the explanation of the Fermi energy tuning.

16	Supplementary note 9	We newly added this note 9 to explain the spin polarization of the AMSs in the coupled JJs with the strong SOI.
17	Supplementary note 10	We newly added this note 10 to follow the previous theoretical studies about the zero energy states emergent in the coupled JJs and multiterminal JJs.
18	Supplementary note 11	We newly added this note 11 to represent the calculated spectra for the single, weakly coupled, and strongly coupled JJs with no SOI.
19	Fig. S7	We slightly changed the colors and the size of the graphs.
20	Fig. S8	We slightly changed the labels for (c) and (d).
21	Fig. S9	We newly added this figure for the supplementary note 9.
22	Fig. S10	We newly added this figure for the supplementary note 10.
23	Fig. S11	We newly added this figure for the supplementary note 11.

Response to Referee #1

Thank you for your comment on our results. We really appreciate that you find the importance of our work. We answer your questions as below. We believe that your questions are adequately addressed, and the improved manuscript deserves publication.

Your comment 1	In the manuscript the authors address an interesting topic : an artificial molecule resulting from two coupled Josephson junctions. The Andreev bound states in each junction play the role of the energy levels of an atom while the overlap among ABS from two different junctions sharing a common electrode changes the energy spectrum of each junction and simulates a molecule. This result is not very surprising by itself but it opens up new perspectives in the field of multi-terminal Josephson junctions which has been recently suggested as an interesting platform for topological matter. The junctions are obtained by coupling two aluminium superconducting electrodes through a In As quantum well. The experiment is performed with care and the results sound correct. The ABS energy depends on the phase difference. In this work, phase bias of the two Josephson junctions is provided by the magnetic flux through two identical superconducting loops. By tunneling spectroscopy, it is shown that when an Andreev molecule forms then the gap closes as a function of the applied magnetic flux. This is the main result. Simulations seem to indicate that this is the case only when spin-orbit coupling (SOC) is included. However the role of SOC in the gap is unclear. For instance, no simulations are shown for a single junction.
Our answer	We thank you for the critical review of our manuscript. Prompted by your kind question, we re-examined our numerical results. We have refined the numerical calculation section including Fig. 3 and added explanations about the spectra in the main text. We have compared the numerical results of a single JJ, weakly coupled JJs, and strongly coupled JJs. The role of SOI in the spectra is to induce spin-splitting of the ABSs. In the literature, the spin-split ABS spectra have been proposed and experimentally reported in the two-terminal JJs [Phys. Rev. X 9, 011010 (2019)]. The spectra can be depicted in Fig. 3(a1). In the case of two-terminal JJs, the spin splitting of the ABSs, namely the Kramers degeneracy lifting is weak and the change of the spectra is not remarkable. However, when the AMSs are formed in the strongly coupled JJs, the level crossing of the formed AMSs appears in $0 <$

	$\phi_L = \phi_R < \pi$ and $\pi < \phi_L = \phi_R < 2\pi$. If the SOI does not exist in the coupled JJs, level crossing of the spin-degenerated AMSs is expected at zero energy and then the level hybridization of the AMSs causes the opening of the SC gap as shown in Fig. S10(c). On the other hand, if the strong SOI exists, the level crossing of the AMSs with the same spins appears at finite energies, while the crossing of the AMSs with the different spins occurs at zero energy as depicted in Fig. 3(c2). The latter crossing is protected because the two AMSs with different spins are the time-reversal invariant pairs, resulting in the SC gap closing. Therefore, to explain our SC gap closing results in the experiments, the SOI role is important. To guarantee the level crossing behavior, we analyzed wavefunctions and spin polarization of ABSs in the vicinity of level crossing points in Fig. S9. The logscale plots of the positive ABS energy indicate the behavior that the positive ABS touches on the zero energy. The negative ABSs also show such zero energy touching because of the particle hole symmetry of the BdG equation. We calculated the spin polarization of the positive and negative ABSs and found that just before and after the zero energy touching points, the spin polarizations of the positive and negative ABSs are exchanged. These support that the positive spin-split ABS becomes the negative ABS and vice versa when the phase differences are swept through the zero energy touching points and namely the presence of zero energy states in the coupled JJs in the numerical calculation. We added the discussion in the main text and the more detailed discussion in the supplementary. (Revision list: #7,10,15,16,17,18,19,20,21, and 22)
Your comment 2	Without a clear and convincing explanation on why SOC is essential for gap closing I can not recommend publication. A quasi quantitative analysis supporting such a conclusion is also needed. This is important also because recent work on very similar devices (arXiv:2302.14535) seems not to show such a gap closing. On the same foot it would be interesting to show that the gap closing disappears when the width of the electrode common to both Josephson junctions becomes larger than the superconducting coherence length.
Our answer	Thank you very much for your critical suggestion. As we explained in our answer to your question 1, we added our new calculation results and conceptual images to explain the experimentally and numerically observed spectra in Figs. 2 and 3. The role of SOI is to cause the spin-splitting of the ABSs. When the AMSs with the SOI are formed with sufficient strong coherent coupling, level crossing of the AMSs with the different spins occurs at the zero energy. This is an essential role of SOI on the SC gap closing.

	As for the comparison with arXiv:2302.14535, first of all, the previous theory has already proposed that the three-terminal JJs with the strong SOI also indicate the SC gap closing [Phys. Rev. B 90, 155450 (2014)]. Therefore, the experimental data can also indicate the SC gap closing. However, in the arXiv they performed tunnel spectroscopy using an SC lead while we used a normal metal lead (2DEG in the InAs quantum well). This difference significantly affects the measured spectra around the zero energy because in the tunnel spectroscopy using the SC lead, the obtained data includes the SC gap of the SC lead. Therefore, the ABSs in the positive and negative energies are separated with twice of the SC gap in the raw data. For example, they show their ABS spectra in Figs. 2a,b,c, which includes the SC gap of the lead in the positive and negative energy sides. Therefore, the gap looks open at a glance. However, in their Fig. 2f or g for example, the AMS curves touch at the SC gap energy ($V = -\Delta/e$). If they correctly subtract the SC gap derived from the lead, this behavior should correspond to the level crossing at zero energy, namely the SC gap closing. In this sense, our results do not conflict with the results in arXiv:2302.14535. We performed the numerical calculation of the weakly coupled JJs whose center SC lead is 1000 nm length comparable to the Aluminum coherence length in Fig. 3(b1). Note that we used 160 nm length for the center SC lead in the calculation of the strongly coupled JJs. When the level hybridization of the ABSs from J1L and J1R is weak (level hybridization is smaller than the SC gap in the single J1L or J1R around the phase difference of π), no level crossing is expected at zero energy as conceptually depicted in Fig. 3(b2). We added this discussion in the main and supplementary. (Revision list: #7 and 10)
Your comment 3	Finally a more focused introduction on the consequences of the SOC on the ABS spectrum would help the reader in a better understanding of the main result. Some sentences (see for instance : « the induced SC gap is defined as the ABSs with no momentum along the junction and the ABSs with finite momentum fill the states outer of the gap ») sound unclear.
Our answer	Thank you for your kind suggestion. We added a paragraph of the background about the ABS spectrum with the SOI and refined the paragraph about the multiterminal JJs' background. In addition, we improved our writing including the sentence which you pointed out as an instance. (Revision list: #2, 3, and 6)

Response to Referee #2

Thank you for your comment on our results. We really appreciate that your professional comments and suggestions which help to improve our manuscript. We refined our manuscript based on your comment and suggestions. We answered to your questions as below. We believe that your questions are adequately addressed, and the improved manuscript deserves publication.

Your comment 1	In “Phase-dependent Andreev molecules and superconducting gap closing in coherently coupled Josephson junctions” Matsuo et. al report tunnel spectroscopy of two coupled Josephson junctions. The authors perform individual spectroscopy of each junction with the other one depleted, effectively realizing an ordinary two-terminal junction, and then with both junctions activated to study the coupled case. Some modeling of spin-orbit coupled Andreev bound states is provided. The authors show how changing the gate voltage on one junction continuously alters the spectroscopy in another junction. Particular attention is paid to an experimentally observed gap closure feature.
Our answer	We thank you for the critical review of our manuscript.
Your comment 2	The main claims are that the two junctions are coherently coupled and that the superconducting gap closes due to the formation of an Andreev molecule. It does appear that there is a coherent interaction between Andreev states in the two junctions. However, I am concerned the discussion surrounding the gap closure. In particular, the authors relate their observation of a gap closure to the presence of spin-orbit interaction in the junction. I do not think this claim is actually supported by the data. First, I would like to emphasize that as an experimental observation, the gap closure is completely clear, as shown in Figs. 2g,2h. I have no problem with this. The problem lies with the discussion of these features and their interpretations.
Our answer	Thank you for your comment. We agree with your opinion that the discussion in the previous version is insufficient and our claim about the SOI in the previous manuscript is unclear. To make interpretation on our data clearer, we performed the additional numerical calculation and added our explanation about the mechanism of the SC gap closing as we wrote in our answer to your question 3. We found that both the SOI and the AMS formation are important to cause the SC gap closing

	theoretically because the zero energy states consistent with the SC gap closing appear when the spin-split AMSs with the different spins cross at zero energy. From the experiments and the comparison with the calculations, we can say that the obtained SC gap closing is a feature possibly emergent in the AMS spectra with the SOI. We refined our manuscript following this direction. (Revision list: #4, 7, and 10)
Your comment 3	Although the authors state “Observation of this SC gap closing is an important step to realize and control more exotic SC phenomena predicted in the multiterminal JJs including Weyl fermion physics” I do not see a completely clear claim in the main text as to the origin of the gap closure. The authors show some numerical simulations and discuss related theories of topological transitions in planar multi-terminal junctions or time-reversal symmetry breaking in small multi-terminal junctions.
Our answer	Thank you for the valuable suggestion. As for the origin of the SC gap closing, we added several conceptual results in Fig. 3 to explain the mechanism of the zero energy crossing behavior owing to the forming of AMS and the SOI. We have compared the numerical results of a single JJ, weakly coupled JJs, and strongly coupled JJs. The role of SOI in the spectra is to induce spin-splitting of the ABSs. In the literature, the spin-split ABS spectra have been proposed and experimentally reported in the two-terminal JJs [Phys. Rev. X 9, 011010 (2019)]. The spectra can be depicted in Fig. 3(a2). In the case of two-terminal JJs, the spin splitting of the ABSs, namely the Kramers degeneracy lifting is weak and the change of the spectra is not remarkable. However, when the AMSs are formed in the strongly coupled JJs, the level crossing of the formed AMSs appears in $0 < \phi_L = \phi_R < \pi$ and $\pi < \phi_L = \phi_R < 2\pi$. If the SOI does not exist in the coupled JJs, level crossing of the spin-degenerated AMSs is expected at zero energy and then the level hybridization of the AMSs causes the opening of the SC gap as shown in Fig. S10(c). On the other hand, if the strong SOI exists, the level crossing of the AMSs with the same spins appears at finite energies, while the crossing of the AMSs with the different spins occurs at zero energy as depicted in Fig. 3(c2). The latter crossing is protected because the two AMSs with different spins are the time-reversal invariant pairs, resulting in the SC gap closing. Therefore, to explain our SC gap closing results in the experiments, the SOI role is important. To guarantee the level crossing behavior, we analyzed wavefunctions and spin polarization of ABSs in the vicinity of level crossing points in Fig. S9. The logscale plots of the positive ABS energy indicate the behavior that the positive ABS

	touches on the zero energy. The negative ABSs also show such zero energy touching because of the particle hole symmetry of the BdG equation. We calculated the spin polarization of the positive and negative ABSs and found that just before and after the zero energy touching points, the spin polarizations of the positive and negative ABSs are exchanged. These support that the positive spin-split ABS becomes the negative ABS and vice versa when the phase differences are swept through the zero energy touching points and namely the presence of zero energy states in the coupled JJs in the numerical calculation. As for the topological transition, we re-constructed our explanation in the main text. In the previous theory of the SC gap closing in the multiterminal JJs (Physical Review B 90, (2014)), they theoretically derived the allowed regions for the emergent zero energy states in the multiterminal JJs with strong SOI. When the multiterminal JJs with strong SOI hold the zero energy states, the zero energy states appear in the region that the three SC phases surround the origin of the circle. The SC gap closing points in our experimental and numerical results are included in the theoretically proposed regions, which we explained in supplementary note 10. Then, we move on the discussion of Phys. Rev. B 106, L241405 (2022). When the zero energy states appear in the coupled planar JJs with strong SOI (a kind of the multiterminal JJs), the finite region encircled by the zero energy can exist. In the PRB paper they claim that the finite regions encircled by the zero energy states are topological SC states classified into the Altland-Zirnbauer topological symmetry class D. In this sense, we consider that our results are related to the topological SC state (Majorana zero modes) and can be a first step towards the demonstration. We removed our previous claim about the Weyl fermions because the Weyl physics is less relevant than the topological SC state. We added the above discussion and explanation in our main and supplementary. (Revision list: #7,8,10,15,16,17,18,19,20,21, and 22)
Your comment 4	My reading is that the authors are saying that, based on comparing the data with numerical simulations in Fig. 3a,3b, the data support the spin-orbit coupled case in 3b. However, I do not believe the data in reality distinguish between the small gap scenario (3a) and "gapless" scenario (3b). I use quotes because actually both cases are gapped in the provided simulations. It would indeed be interesting if there was evidence that spin-orbit is making the gap close, but the data the authors present appears entirely consistent with both cases.

Our answer	Thank you very much for an important suggestion to polish our discussions. First, we added some numerical calculation and discussion to insist the SC gap closing (zero energy states produced by level crossing of the AMSs with the different spins at the zero energy). As we already wrote in our answer to your question 3, we analyzed the wavefunctions and the spin polarization of the ABSs in the vicinity of level crossing points in Figs. 3(c1) and (d). The calculated spin polarization exhibits that the spin-split AMS level changes from positive to negative and vice versa when the phase differences are swept through the crossing points. This behavior guarantees that the crossing points at zero energy are protected because the ABSs with the different spins are time-reversal invariant pairs. Therefore, these results clearly indicate the SC gap closing in the coupled JJs with strong SOI as theoretically predicted. Every experimental setup about the spectroscopy has finite energy resolution. Therefore, we cannot distinguish the SC gap closing from the SC gap smaller than the energy resolution. However, at least, the previous theoretical studies and our results of numerical calculation both suggest that the SC gap closing behavior can appear in the coupled JJs with the strong SOI. Herein, our interpretation of the experimental data as the SC gap closing is not inconsistent with the theoretical expectation. Then we think it is possible to insist that the SC gap closing expected in the coupled JJs holding the SOI is observed experimentally. As we wrote in our answer to your question 2, we refined our manuscript to change our claim from the previous (“ we found that the SOI is necessary to make the SC gap closing”) to the current version. We added the above discussion. (Revision list: #7,16, and 21)
Your comment 5	A related point of confusion is what regime the authors actually claim this gap closure corresponds too. Since the fluxes are equal I presume they do not intend to claim time-reversal symmetry breaking in the junctions. Is the idea that this is akin to the fine-tuned class DIII point of Ref. [30], which is indeed gapless after ensemble averaging?
Our answer	Thank you for your comment and we are sorry that our previous explanation is insufficient. Phys. Rev. B 106, L241405 (2022) explains the symmetry class of the considered finite region encircled by the zero energy states. The symmetry class is D in the Altland-Zirnbauer topological symmetry classification. We added this explanation in the main text. (Revision list: #7, 13, and 17)
Your comment	I am not enthusiastic about this manuscript appearing in nature communications, but if the authors wish to respond I would like to see a much more clear statement

6	about what claims are being made surrounding the gap closure, and a careful argument of why the data agree better with 3(b) than 3(a).
Our answer	In the refined manuscript, we carefully explain the origin of the AMS spectra experimentally and numerically obtained and clearly provide the SOI roles on the spectra and the SC gap closing. In addition, theoretical explanation including the contents of the previous theoretical efforts about the finite region encircled by the zero energy states is given in the main and supplementary. We believe these improvements of our manuscript sufficiently make our claim and the importance clearer. We expect that you find our revised manuscript is valuable for the publication.
Your comment 7	If I take away the discussion of gap closure, there remains perfectly fine evidence that an Andreev molecule has been realized. But, as the authors point out in their literature review, other Andreev molecules have also appeared, and this case does not appear particularly clean or tunable.
Our answer	Thank you for your comment. In the literature, some groups reported the AMS evidence in their tunnel spectroscopy. Nat Commun 8, 585 (2017) reported the AMSs (hybridized Andreev levels) in a two-terminal JJ of the double quantum dot in a single nanowire (S-QD-QD-S). In this case, the coherent coupling of the Andreev levels formed in the respective quantum dots coupled to the single SC leads (S-QD) occurs through the tunnel barrier formed by the electrical gating. The AMS study in Nano Lett. 21, 7929 (2021) performed the spectroscopy of the single SC leads contacted to two quantum dots using the other SC lead. In this case, also the Andreev levels formed in the respective quantum dots are hybridized through the shared SC lead. arXiv:2111.00651 reported the AMS in the very similar structures with Nat Commun 8, 585 (2017). All the reports have not used the Josephson junctions but the Andreev levels formed in the S-QD as the ingredients of the AMSs. Therefore, their results lack evidence of “coherence” in the AMSs. In our results we clearly give the phase-dependent spectra by means of the SC quantum interference. Our results provide the microscopic evidence of the coherent coupling of the ABSs in two JJs through the shared SC lead in which the nonlocal SC transport phenomena invoked by the phase control have been demonstrated recently. In this sense, our observation of the phase-dependent AMS is important even without the claim of the SC gap closing. These are reflected on the sentences of “although there are experimental reports of the AMS signatures formed in SC junctions other than JJs [31–33].” in the main text.

Your comment 8	As a minor comment, in the Fig. 1(h) caption “GR vs. V measured for the coupled JIL.” I think the authors intend JIL -> JIR?
Our answer	Thank you for your kind remark. We checked the Fig. 2h and found the mistake. We corrected the typo from JIL to JIR. (Revision list: #13)

Response to Referee #3

Thank you for your comment on our results. We really appreciate that you find the importance of our work. We answer your questions as below. We believe that your questions are adequately addressed, and the improved manuscript deserves publication.

Your comment 1	I have reviewed the manuscript entitled "Phase-dependent Andreev molecules and superconducting gap closing in coherently coupled Josephson junctions" which reports primarily experimental results on tunneling spectroscopy in a three-terminal setup comprising two coupled Josephson junctions. The reported experimental findings appear to be solid and reproducible with relevance for the readers of Nature Communications. I have, however, two reservations to be addressed before I can recommend the manuscript for publication.
Our answer	We thank you for the critical review of our manuscript.
Your comment 2	First, I find the language of the manuscript somewhat clumsy with parts which are not really clear such as, e.g., the sentence "The induced gap..." on lines 120 and 121 or "oscillates TO B" on l. 422. General check of English would be beneficial.
Our answer	Thank you for your remark and sorry for the confusion. We asked our colleague who is a native English speaker from UK. We believe that our English writing is now sufficiently improved. (Revision list: #1,5,12, and 13)
Your comment 3	Furthermore, I would recommend to explicitly mention in the caption of Fig. 2 the three colors above the upper panels corresponding to the lines in the lower panels. This is mentioned in the main text only and is thus somewhat missing in the caption itself.
Our answer	Thank you for your recommendation. We added an explanation of the colors in the Fig 2 caption. (Revision list: #9)
Your comment 4	My last question about the presentation concerns the Fig. 4. Is really the mag. field axis in panel 4c 10-times longer than in panels 4a? The gap closing within the range of 0.1 mT of panels 4a is not seen at all in panel 4c - is it deliberate and, if so, why?
Our answer	Thank you for your comment. This label for the horizontal axis in Fig. 4c was incorrect and the axis range should be the same as the panels of Fig. 4(a). We corrected the label of Fig. 4c. The gap closing in Fig. 4c occurs at the same B points in Fig. 4a. (Revision list: #11)
Your	More serious problems I see in the modelling part of the manuscript. Even though

comment 5	I do understand a very short space for any detailed description of the theoretical concepts, I haven't got any specific image of the mechanism involved in the observed effects. The whole section on "Numerical calculation of the AMS..." is just a series of vague hints on various phenomena and list of citations without any more specific conclusion (in my view). On the other hand, the calculations in the supplement 7 are for a very specific model without any context (no references at all). Altogether, neither the main text nor the supplementary note provide a coherent and/or conclusive picture of the expected microscopic mechanism, which I view as a serious shortcoming of the present manuscript. I strongly suggest the authors to amend this situation.
Our answer	We thank you for the critical comment and apologize our lack of the model explanation. We have refined the numerical calculation section in Fig. 3 and added explanations about the spectra in the main text. We have compared the numerical results of a single JJ, weakly coupled JJs, and strongly coupled JJs. The role of SOI in the spectra is a spin-splitting of the ABSs. In the literature, the spin-split ABS spectra have been proposed and experimentally reported in the two-terminal JJs [Phys. Rev. X 9, 011010 (2019)]. The spectra can be depicted in Fig. 3(a1). In the case of two-terminal JJs, the spin splitting of the ABSs, namely the Kramers degeneracy lifting is weak and the change of the spectra is not remarkable. However, when the AMSs are formed in the strongly coupled JJs, the level crossing of the formed AMSs in $0 < \phi_L = \phi_R < \pi$ and $\pi < \phi_L = \phi_R < 2\pi$. If the SOI does not exist in the coupled JJs, level crossing of the spin-degenerated AMSs is expected at zero energy and then the level hybridization of the AMSs causes the SC gap as shown in Fig. S11(c). On the other hand, if the strong SOI exist, the level crossing of the AMSs with the same spins at finite energies while the crossing of the AMSs with the different spins at zero energy as depicted in Fig. 3(c2). The latter crossing is protected because the two AMSs with the different spins are the time-reversal invariant pairs, resulting in the SC gap closing. Therefore, to explain our SC gap closing results in the experiments, the SOI role is important. To guarantee the level crossing behavior, we analyzed wavefunctions and spin polarization of ABSs in the vicinity of level crossing points in Fig. S9. The logscale plots of the positive ABS energy indicate the behavior that the positive ABS touches on the zero energy. The negative ABSs also show such zero energy touching because of the particle hole symmetry of the BdG equation. We calculated

the spin polarization of the positive and negative ABSs and found that just before and after the zero energy touching points, the spin polarizations of positive and negative ABSs are exchanged. These support that the positive spin-split ABS becomes the negative ABS and vice versa when the phase differences are swept through the zero energy touching points and namely the presence of zero energy states in the coupled JJs in the numerical calculation.

In addition, we added a detailed explanation of our model in supplementary information and the detailed related theory background in the main and supplementary information.

We added the discussion in the main text and the more detailed discussion in the supplementary. (Revision list: #7,10,14,15,16,17,18,19,20,21, 22, and 23)

REVIEWER COMMENTS

Reviewer #1 (Remarks to the Author):

The manuscript has been completely revised and it can now be considered for publication in Nature Communications. The authors addressed most of the criticisms I brought up. However one point should be clarified before publication. The simulations show indeed that the gap closing appears only for strongly coupled AMS in presence of SOI. It would be interesting to understand quantitatively the effect of SOI on AMS. Is the splitting induced by the SOI in InAs large enough to ensure the gap closing ?

Reviewer #2 (Remarks to the Author):

The authors have significantly revised the manuscript and I am grateful for their efforts. It is now much more clear what claims they are making, and what their framework is for comparing with theory.

In my view, it is still not plausible that the authors have resolved any role of spin-orbit coupling. Taking the no spin-orbit case (Fig. S11c) as a null hypothesis, I see that their experiment would need to resolve gaps at the level of 0.1Δ to distinguish the spin-orbit and no spin-orbit case. This is well beyond the current tunnel probe, so I do not see any evidence against the null (no spin-orbit) hypothesis.

Based on this logic, I don't see the spin-orbit claim as strong enough to have a central role. The authors could remedy this by simply stating that their tunnel probe cannot plausibly resolve the difference between the two cases. I do not have a problem with notifying readers that in theory gap closure is in theory related to breaking of Kramers degeneracy, although it probably should not be in the abstract.

Aside from this point I find the manuscript is now reasonably clear and I don't have other fundamental concerns.

Revision List

We uploaded a supporting file (main_v10_compared) in which we highlighted the revisions with red and indicated the line numbers.

#	Revision on the main text	How to revise
1	L27	We toned down our claim about the SOI causing the SC gap closing in the abstract.
2	L204-211	We refined the discussion of the comparison between the experimental data and the calculated results. We added the other possible scenario to explain our data with negligible SOI.
3	L262	We removed our strong claim about the SC gap closing originating from the SOI.
4	Reference list	We added related references (#24,26,27).
5		We corrected some typos.

We have no change in our supplementary information.

Response to Referee #1

Thank you for your comment on our results and for highly evaluating our revised manuscript.

We answer your question as below.

Your comment 1	The manuscript has been completely revised and it can now be considered for publication in Nature Communications. The authors addressed most of the criticisms I brought up. However one point should be clarified before publication. The simulations show indeed that the gap closing appears only for strongly coupled AMS in presence of SOI. It would be interesting to understand quantitatively the effect of SOI on AMS. Is the splitting induced by the SOI in InAs large enough to ensure the gap closing ?
Our answer	Thank you for your consideration of our manuscript. At least in our numerical calculation, we used the typical SOI value reported in InAs quantum wells ($\alpha \approx 4.14 \times 10^{-11}$ eV · m,) as written in supplementary note 7. Consequently, we found the splitting of AMS in our numerically obtained spectra, for example in Fig. 3(c1). Therefore, it is not strange that the splitting is large enough to ensure the gap closing. However, as reviewer #2 suggested, there seems an additional scenario that the small SC gap with the negligible SOI as found in Fig. S11(c) is smeared out due to the thermal effect or our measurement resolution. We put a remark about this in the main manuscript (#1,2,3).

Response to Referee #2

Thank you for your comment on our results and for highly evaluating our revised manuscript. We followed your suggestion as below. We believe that the improved manuscript deserves publication.

Your comment 1	The authors have significantly revised the manuscript and I am grateful for their efforts. It is now much more clear what claims they are making, and what their framework is for comparing with theory. In my view, it is still not plausible that the authors have resolved any role of spin-orbit coupling. Taking the no spin-orbit case (Fig. S11c) as a null hypothesis, I see that their experiment would need to resolve gaps at the level of $0.1 \text{ } \Delta$ to distinguish the spin-orbit and no spin-orbit case. This is well beyond the current tunnel probe, so I do not see any evidence against the null (no spin-orbit) hypothesis. Based on this logic, I don't see the spin-orbit claim as strong enough to have a central role. The authors could remedy this by simply stating that their tunnel probe cannot plausibly resolve the difference between the two cases. I do not have a problem with notifying readers that in theory gap closure is in theory related to breaking of Kramers degeneracy, although it probably should not be in the abstract. Aside from this point I find the manuscript is now reasonably clear and I don't have other fundamental concerns.
Our answer	Thank you for your suggestion. We admitted that the experimental results can be assigned to a scenario in which the small SC gap with the weak or negligible SOI in our device in Fig. S11(c) is smeared out due to the thermal effect or our measurement resolution. Therefore, following your suggestion, we toned down our claim of the SC gap closing derived from the spin-orbit interaction in the main manuscript. (#1,2,3)

REVIEWERS' COMMENTS

Reviewer #1 (Remarks to the Author):

The authors clarified the role of SOI. I have no further suggestions to improve the quality of the manuscript that can, I think, be published as it is.

Reviewer #2 (Remarks to the Author):

The SOI language has indeed been toned down, and this could reasonably be published as is.

I must mention that the discussion of spin-orbit coupling starting on line 205 is still a bit problematic:

"This discrepancy between the experimental results and theoretical calculation can be attributed to the effect of our measurement resolution and thermal smearing. Due to the effect, the two SC gap closing points in the strong SOI case depicted in Fig. 3(c1) can overlap or the small SC gap in the weak or negligible SOI case (supplementary note 11 and Fig. S11) is smeared out. Therefore, we conclude that the experimentally observed Andreev spectra in Fig. 2 reflect the phase-dependent AMSs between the two 210 JJs holding the strong SOIs."

The last sentence concludes that the experimental observations reflect the role of strong spin-orbit coupling. It is a non sequitur. The previous sentence indicates that the results are consistent with both the strong and weak spin-orbit coupling cases once reasonable levels of smearing are taken into account. It follows that the data do not reflect strong SOI and more than they reflect weak SOI.

Revision List

We highlighted the revisions with red and indicated the line numbers.

#	Revision on the main text	How to revise
1	L209	We revised the text that the reviewer #2 pointed out.

We have no change in our supplementary information.

Response to Referee #2

Your comment 1	The SOI language has indeed been toned down, and this could reasonably be published as is. I must mention that the discussion of spin-orbit coupling starting on line 205 is still a bit problematic: "This discrepancy between the experimental results and theoretical calculation can be attributed to the effect of our measurement resolution and thermal smearing. Due to the effect, the two SC gap closing points in the strong SOI case depicted in Fig. 3(c1) can overlap or the small SC gap in the weak or negligible SOI case (supplementary note 11 and Fig. S11) is smeared out. Therefore, we conclude that the experimentally observed Andreev spectra in Fig. 2 reflect the phase-dependent AMSs between the two 210 JJs holding the strong SOIs." The last sentence concludes that the experimental observations reflect the role of strong spin-orbit coupling. It is a non sequitur. The previous sentence indicates that the results are consistent with both the strong and weak spin-orbit coupling cases once reasonable levels of smearing are taken into account. It follows that the data do not reflect strong SOI and more than they reflect weak SOI.
Our answer	Thank you for your suggestion. We revised our text that you pointed out.